# Structural filtering of functional data offered discriminative features for autism spectrum disorder

**Alireza Talesh Jafadideh**, **Babak Mohammadzadeh Asl** *

Department of Biomedical Engineering, Tarbiat Modares University, Tehran, Iran

* babakmasl@modares.ac.ir

**Data Availability Statement:** All rfMRI and DTI files are available from the ABIDE II database with URLs (needing registering in NITRC): http://fcon_1000.projects.nitrc.org/indi/abide/abide_II.html and

## Abstract

This study attempted to answer the question, "Can filtering the functional data through the frequency bands of the structural graph provide data with valuable features which are not valuable in unfiltered data"?. The valuable features discriminate between autism spectrum disorder (ASD) and typically control (TC) groups. The resting-state fMRI data was passed through the structural graph's low, middle, and high-frequency band (LFB, MFB, and HFB) filters to answer the posed question. The structural graph was computed using the diffusion tensor imaging data. Then, the global metrics of functional graphs and metrics of functional triadic interactions were computed for filtered and unfiltered rfMRI data. Compared to TCs, ASDs had significantly higher clustering coefficients in the MFB, higher efficiencies and strengths in the MFB and HFB, and lower small-world propensity in the HFB. These results show over-connectivity, more global integration, and decreased local specialization in ASDs compared to TCs. Triadic analysis showed that the numbers of unbalanced triads were significantly lower for ASDs in the MFB. This finding may indicate the reason for restricted and repetitive behavior in ASDs. Also, in the MFB and HFB, the numbers of balanced triads and the energies of triadic interactions were significantly higher and lower for ASDs, respectively. These findings may reflect the disruption of the optimum balance between functional integration and specialization. There was no significant difference between ASDs and TCs when using the unfiltered data. All of these results demonstrated that significant differences between ASDs and TCs existed in the MFB and HFB of the structural graph when analyzing the global metrics of the functional graph and triadic interaction metrics. Also, these results demonstrated that frequency bands of the structural graph could offer significant findings which were not found in the unfiltered data. In conclusion, the results demonstrated the promising perspective of using structural graph frequency bands for attaining discriminative features and new knowledge, especially in the case of ASD.

## Introduction

Autism spectrum disorder (ASD) is a neurological and neurodevelopmental disorder characterized by impaired social communication, and restricted and repetitive interests, behavior,

https://www.nitrc.org/ir/data/projects/ABIDE_II. we used the SDSU and NYU1 datasets of ABIDE II. The link and DOI of author-generated codes are: https://figshare.com/articles/software/MATLAB_Codes/21608031 (DOI 10.6084/m9.figshare.21608031).

**Funding:** The author(s) received no specific funding for this work.

**Competing interests:** The authors have declared that no competing interests exist.

and activities [1]. Because of the rapid prevalence and heterogeneous etiology of ASD, a variety of research has been devoted to finding biomarkers and features discriminating ASDs from typically controls (TCs) [2–5].

One of the popular approaches for finding reliable and quantifiable biomarkers is studying the functional behavior of the brain through resting-state functional magneto resonance imaging (rfMRI) data [6–11]. However, the inherent complexity of the human brain makes its functional behavior study a challenging issue [12]. To handle this complexity, the brain is usually modeled by a graph whose vertices are regions of interest (ROI)s of the brain, and edges represent the functional connectivity (FC) between regions of interest (ROIs) [13, 14]. The most commonly used metrics of the graph can be classified into two groups: local and global metrics [15, 16]. The earlier ones describe the network behavior at the ROI level and offer discriminative features at this level. The latter ones describe the properties governing the entire brain network.

The ASD leads to abnormal FCs [10, 11], which in turn change the graph metrics. Thus, these changes can be revealed and regarded as candidate biomarkers [17]. Di Martino and colleagues [18] found that cortical and subcortical ROIs of ASDs had more connections (degree) than TCs. Redcay E and colleagues [19] reported the greater betweenness centrality, a metric indicating the ROI centrality, for ASDs compared to TCs when studying the right parietal region of the default mode network (DMN). These findings represent the over-connectivity in ASDs compared to TCs [17]. Rudie and colleagues [20] reported decreased local and increased global efficiency in adolescents with ASD. Itahashi and colleagues [21] found reduced characteristic path length and clustering coefficient when studying adult ASDs in comparison to TCs. The results of these two papers show the increased randomness of functional networks for ASDs' brains [20, 22]. Keown and colleagues [22] reported findings at the global level. Their analysis showed globally reduced cohesion but increased dispersion of networks. Cohesion quantified how well ROIs from a normative network grouped within the community structure. Dispersion quantified how distributed ROIs from a network were in the community structure. These results demonstrated impaired network integration and segregation of ASD subjects [22]. Zhou and colleagues performed a small-world network (SWN) analysis using the FC and did not find any differences between ASD and TC children [23]. Altogether, impaired functional organizations of the autistic brain were found in both local and global metrics of the graph [16].

In most of the graph metrics and methods of FC analysis, the dyadic interactions are exploited. However, the human brain is one of the world's most complex networks, and the interaction of two ROIs is not independent of the rest of the ROI-ROI interactions. Thus, a higher level of interactions exists in the brain network, and investigating them can provide new valuable findings. Recently, analyzing such higher-level interactions was carried out by Moradimanesh and colleagues [24]. The authors analyzed the brain triadic interactions to compare ASD and TC groups. Three ROIs and their interactions with each other were considered for analyzing the triadic interactions. The ROIs interactions was FC measured by Pearson correlation. The authors investigated four types of triad interactions, including strongly balanced $T_3$: $(+++)$, weakly balanced $T_1$: $(+--)$, strongly unbalanced $T_2$: $(++-)$, and weakly unbalanced $T_0$: $(---)$. The "+" and "-"are the sign of Pearson FC between ROIs. They found that balanced and unbalanced triads were over-presented and under-presented in both ASD and TC groups, respectively. Also, they observed that the energy of the salience network (SN) and the default mode network (DMN) were lower in ASD, probably indicating the difficulty of flexible behavior.

The brain signals not only change over time but also change over the brain topology. The topology domain is irregular. As a result, the signal changes in the topological domain cannot

be captured by the Fourier transform of the time domain. To address this problem, the field of graph signal processing (GSP) has recently emerged, attempting to develop methods for capturing frequencies of the topological domain [25, 26]. The GSP needs two elements: graph and signal. The graph represents the underlying structure of the signal, and the signal is brain activity mounted on the graph. Using these elements and graph Fourier transform (GFT), one can deal with topological frequencies. The GFT is one of the GSP tools for working with topological frequencies. Recently, the GFT has been used in two studies of autism to classify ASDs and TCs [27, 28]. Brahim and Farrugia used the structural connectivity (SC) matrix of the Human Connectome Project (HCP) to obtain the graph of GSP [27]. This matrix was attained by averaging over 56 TC subjects. The authors used this averaged structure as the underlying structure of all studied ASD and TC subjects. Some statistical metrics including standard deviation (SD), mean, variance, and Kurtosis were computed for each ROI. These calculated metrics were considered as the signal of GSP. These signals and their GFT were used as features. The authors found that the feature with the best classification performance was the GFT of SD. Itani and Thanou used the inverse of the distance between ROIs as weights to construct the graph of GSP [28]. The authors proposed a framework using GFT and spatial filtering method (SFM) to construct a subspace extracting discriminative features between ASDs and TCs. The classification using the decision tree demonstrated the superiority of their proposed method performance compared to other state-of-the-art ones.

Does passing rfMRI data through topological filters offer data with discriminative features between ASDs and TCs? Does filtered data offer discriminative features which are not discriminative in unfiltered data?. This study aimed to answer these questions using some metrics of the graph and triadic interactions as features. These metrics have not been studied in the topological frequency bands. The topological filters were obtained through GFT tool and SC matrix. Each subject had its own SC matrix which was computed using the diffusion tensor imaging (DTI) data. Using GFT, graph frequency modes and, consequently, the topological filters were obtained. Then, these filters were applied to rfMRI data to have data in three independent frequency bands of the structural graph, i.e., low, middle, and high-frequency bands. From now on, these bands are abbreviated as LFB, MFB, and HFB, respectively. For each band, the FC matrix was computed using the filtered data. Also, the FC matrix was computed for unfiltered data, which is named full-frequency band (FFB) data. Some of the graph global metrics and triadic interaction metrics from these FC matrices were computed and compared between ASD and TC groups. For each of the ASD and TC groups, the behavior of metrics was compared between the three aforementioned frequency bands.

The graph of GSP is considered the topology (structure) of the studied system (herein is the brain). It is expected that the topology becomes stable during the time. The stability of SC is much more than FC. As a result, structural connectivity (and not functional connectivity) is used to model the graph of GSP in many neuroimaging studies [27–29]. Hence, in this study, the SC was used to model the graph of GSP.

Many autism studies have described patterns of under- or over-connectivity [17–19] which can be led to abnormal segregation (independent processing in specialized subsystems) and integration (global cooperation between different subsystems) of the resting-state brain [30–32]. In this study, the graph metrics were used to assess functional integration and segregation differences between TC and ASD. Some studies have shown good to excellent repeatability for global metrics, while for local metrics it was more variable and some metrics were found to have locally poor repeatability [33–36]. As a result, the global metrics of the graph were used in this study. In some studies, the **clustering coefficient** (metric for segregation analysis) and **efficiency** (metric for integration analysis) have been reported as the most reproducible metrics of graphs [35, 36]. The **strength** is another metric for studying over/under-connectivity in

autism [17, 37, 38]. Another global metric is the **assortativity coefficient**. Generally, an assortative network is robust against selective ROI failure, and this accelerates the spread of information generated by high-degree ROIs. The excessive assortativity led to poor network performance [39]. Itahashi and colleagues [21] reported reduced assortativity in ASD. This finding is consistent with the view that network organization in the ASD brain shifts toward randomization compared to that in the TC brain. In addition to assortativity, the decreased clustering coefficient and increased efficiency also inform about increased brain network randomness [20, 21]. The more/fewer values of **radius** and **diameter** metrics can show more/less correlated neural activity in spatially distributed ROIs in the ASD group compared to TCs [40]. The **SWN** and small-world propensity (**SWP**) are other well-known metrics of graph tending to display a balance between segregation and integration [15, 41]. In this study, the metrics written in the bold face were employed to see if graph frequency bands can reveal an abnormality in terms of segregation and integration or under/over-connectivity or increased/decreased randomness of ASD brain. Our aim is not to use all of the global metrics of the graph. The metrics mentioned above can be enough for answering our questions raised in the sixth paragraph of this section.

The rest of the paper is organized as follows: Section 2 describes the dataset, preprocessing methods, and brain ROIs and networks. This section is continued by explaining the GFT and graph filtering and then describing the scenarios. The last subsection of this section is devoted to statistical evaluations. Section 3 provides the results. The discussion about the results and conclusions are respectively given in sections 4 and 5.

## Materials and methods

The experimental design has been provided in Fig 1.

### Participants and data acquisition

In this study, the SDSU and NYU1 datasets of ABIDE II database were exploited. In the NYU1 dataset, only 57 subjects (33 ASDs, 24 TCs) had both DTI and rfMRI data. Thus, only these subjects were selected for the next step. Both datasets mainly had male subjects for both ASD and TC groups. Accordingly, only the male subjects were analyzed.

For both datasets, two groups were matched on age, handedness, and head motion while showing significant differences in the social responsiveness scale (SRS). The sample characteristics are listed in Table 1. This information is for subjects whose data survived in preprocessing step. The SRS values of one SDSU subject (ASD) and three NYU1 subjects (ASD = 2, Tc = 1) were missing. The framewise displacement (FD) is explained in subsection 2.2.1.

The data were collected using (GE 3T MR750 scanner, 3T Siemens Allegra) with (eight-channel, eight-channel) head coils. High-resolution structural images were acquired according to (SPGR, TFL) standards' T1-weighted sequences and with TR/TEs of (8.136/3.172 ms, 3.25/3.25 ms), flip angles of (8˚, 7˚), fields of view (FOVs) of ($256\times256$ mm$^2$, $256\times256$ mm$^2$), resolutions of (1 mm$^3$, 1.73 mm$^3$), and numbers of slices of (172, 128). The rfMRI datasets were collected using standard (gradient echo-planar imaging (EPI), EPI) sequences with TR/TEs of (2000/30 ms, 2000/15 ms), flip angles of (90˚, 90˚), FOVs of ($220\times220$ mm$^2$, $240\times240$ mm$^2$), matrix sizes of ($64\times64$, $80\times80$), pixel spacing sizes of ($3.4375\times3.4375$ mm$^2$, $3\times3$ mm$^2$), slice thicknesses of (3.4 mm, 4 mm), slice gaps of (0 mm, 0 mm), axial slices of (42, 33), and total volumes of (180, 180). The DTI protocols were based on EPI sequences. The DTI images consisted of (61, 64) weighted diffusion scans with $b$ values of (1000 sec/mm$^2$, 1000 sec/mm$^2$) and one unweighted diffusion scan with $b$ values of (0 sec/mm$^2$, 0 sec/mm$^2$). The other parameters of data recording included TR/TEs of (8500/78 ms, 5200/78 ms), FOVs of ($128\times128$ mm$^2$,

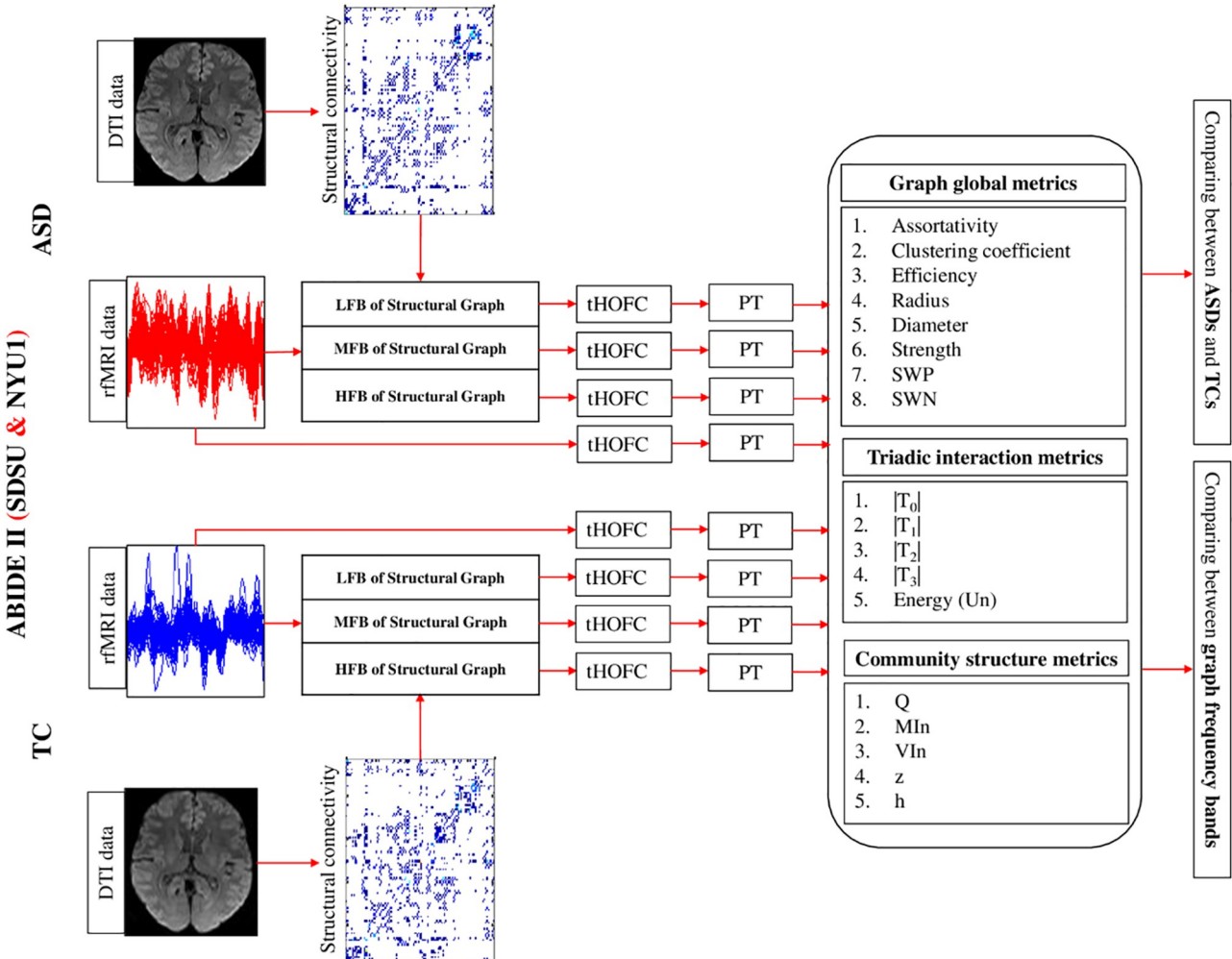

**Fig 1. Overview of pipeline steps and study design.** The rfMRI and DTI data of SDSU and NYU1 datasets of ABIDE II were used in this study. Using the DTI data, the structural connectivity and frequency bands of the structural graph were computed for each subject. The rfMRI data was passed through the structural graph's LFB, MFB, and HFB filters. The FC matrix was computed for filtered and unfiltered data through tHOFC method. Thus, four FC matrices were calculated for each subject. Then, proportional thresholding was applied on FC matrices to remove weak connections. For each FC matrix, the global metrics of the graph and metrics of triadic interactions were computed. Finally, these metrics were compared between ASD and TC groups. Also, for a given group, the metrics results in the LFB, MFB, and HFB were compared. The results of Community structure metrics are provided in the S1 File.

192×192 mm$^2$), slice in-place resolutions of (1.875×1.875 mm$^2$, 3×3 mm$^2$), slice thicknesses of (2 mm, 3 mm), and the numbers of axial slices of (68, 50).

## Approval for human experiments

For each of the included data repositories NYU and SDSU, the following Institutional Review Boards (IRB) and ethics committees have approved the experiments. Data acquisition procedures were following their guidelines and regulations, respectively: (1) Institutional Review Board Operations at NYU, (2) San Diego State University's Human Research Protection Program (HRPP). Furthermore, all experiments followed HIPAA guidelines and 1000 Functional Connectomics Project/INDI protocols; all data were fully anonymized with no protected health information, and legal guardians of all participants signed informed consent.

**Table 1. Sample characteristics.**

| | | ASD | TC | Statistic | *p* |
|---|---|---|---|---|---|
| SDSU and NYU1 | Age (year) | 11.19±4.7 | 11.33±3.35 | $t(96) = -0.17$ | 0.865 |
| | | [5.32–26.6] | [5.88–17.7] | | |
| | Sex | 53 Male | 45 Male | - | - |
| | Handedness | 39 Right | 39 Right | $X^2(1) = 0.29$ | 0.53 |
| | SRS total | 82.56±9.41 | 42.6±5.37 | $t(92) = 17.54$ | <1e-12 |
| | | [42–107] | [34–59] | | |
| | rfMRI FD (mm) | 0.04±0.03 | 0.04±0.03 | $t(96) = 1$ | 0.32 |
| | | [0.01–0.14] | [0.01–0.11] | | |
| | # of rfMRI time points with FD>0.5mm | 0.9±2.5 | 0.23±0.87 | $t(96) = 0.83$ | 0.41 |
| | | [0–11] | [0–5] | | |
| | DTI FD (mm) | 0.27±0.09 | 0.25±0.08 | $t(96) = -0.2$ | 0.84 |
| | | [0.11–0.52] | [0.15–0.61] | | |
| | # of DTI time points with FD>0.5mm | 4.9±5.2 | 3.91±5.38 | $t(96) = 0.63$ | 0.53 |
| | | [0–16] | [0–30] | | |

## Data preprocessing

**rfMRI data preprocessing and time series extraction.** The SPM (http://www.fil.ion.ucl. ac.uk/spm/) and AFNI (https://afni.nimh.nih.gov) software packages and personal codes were used for preprocessing the rfMRI data. To prevent T1-equilibration effects, the first five volumes were ignored. The data were corrected for slice-timing by considering the middle slice as a reference, motion artifact. The data were despiked through the AFNI's 3dDespike function. Then, the spm_coreg function was applied to coregister the T1 image and functional images. The images warped to the standard Montreal Neurological Institute (MNI) space using a template created by nonlinear registration of 152 T1-weighted images. A Gaussian kernel with a 5-mm full width at half maximum (FWHM = 5 mm) was employed for spatial smoothing.

For more noise reduction of data, six motion parameters and their first temporal derivatives and one principal component of cerebrospinal fluid (CSF) and white matter (WM) signals were considered 14 confounds and regressed out from all voxels' time series. CSF and WM principle components explained more than 99% of time series variance and were considered confounds [42]. Then, removing the linear, quadratic, and cubic trends from the time series and applying a band-pass fifth-order Butterworth filter (0.01–0.1 Hz) on the time series were carried out.

To detect the volumes corrupted by motion artifacts, the FD was computed. The FD was defined as the square root of the sum of squares of differences existing in motion parameters of two consecutive time points. The head was considered a sphere with a radius of 50mm to change the values of rotational parameters from radian to millimeter [43]. The time points with FD >0.5 mm and the two subsequent time points were removed [10]. After censoring, as with Mash and colleagues [10], the time series fragments with less than 10 consecutive time points were removed. After censoring, the data was used if more than %75 of its time points were preserved from censoring. The TC subject with ID = 28852 of the SDSU dataset was rejected. The number of subjects of ASD and TC without censoring time points was 42 and 36, respectively ($\chi^2(1) = 0.0009$, $p = 0.97$). There was no significant difference between ASD and TC in terms of the number of censored time points (t(94) = 0.85, $p = 0.39$). Given that the maximum number of censored time points was 20 and the first five volumes were removed to prevent from T1-equilibrium effect, the 155 time points were used for further analysis.

**DTI preprocessing and structural connectivity matrix.** Preprocessing of DTI data and computation of structural connectivity matrix were carried out using the ExploreDTI software [44]. Firstly, data preparation and quality assessment were performed in ExploreDTI. After that, diffusion data were corrected for subjects' motion and Eddy currents. Then, deterministic fiber tracking was performed for tractography. The results of tractography and ROIs of the studied atlas were employed to calculate the structural connectivity matrix. The structural connectivity between two ROIs was defined as the number of tracts between them. In the next step, each matrix row was divided by the sum of its elements. By doing so, the entry value (i,j) expressed the connectivity probability from region i to region j, which was not equal to that of region *j* to region *i*. Thus, the connectivity matrix, also called the adjacency matrix **A**, was not symmetric. As the final step, the symmetric adjacency matrix **A'** was obtained as **A'** = (**A** + **A**$^{\mathbf{T}}$)/2 where.$^{\mathrm{T}}$ denotes the transpose operator. The FDs of DTI data are summarized in Table 1. In ExploreDTI, the REKINDLE (Robust Extraction of Kurtosis INDices with Linear Estimation) [45] detects and removes the outliers.

## Brain ROIs and networks

The Schaefer atlas [46] with 100 ROIs was employed in this study. This atlas is purely based on functional time series and respects the boundaries of 7 Yeo-Krienen functional atlas [47]. For NYU1, one ASD subject was rejected. More than 10% of this subject's ROIs had no structural connections with the rest of the ROIs. By averaging the time series of ROI voxels, one signal was obtained for each ROI. The results for the Schaefer atlas with 200 ROIs are reported in the S1 File.

## Graph frequency bands (GFBs)

The frequency content of the graph signal is defined according to the signal changes across connected vertices at a given time point. In low frequency, connected vertices show similar signals (representing alignment). In high frequency, the variability of the connected vertices signals is high compared to each other (representing liberality). In liberality, the vertices (brain ROIs) show less respect for their underlying connectivity structure. By approaching from low frequency to high frequency, the graph signal behavior changes from alignment to liberality (Fig 2).

The graph frequencies are defined using the combinatorial Laplacian matrix $\mathbf{L} \in \mathbb{R}^{N \times N}$, [25], as follows:

$$\boldsymbol{L} = \boldsymbol{D} - \boldsymbol{A} \tag{1}$$

where **D** is a diagonal matrix, and its $k^{\mathrm{th}}$ diagonal element represents the degree of $k^{\mathrm{th}}$ vertex i.e., $\mathbf{D}_{kk} = \sum_{j=1}^{N} \mathbf{A}_{kj}$. The eigendecomposition of **L** provides the **V** and **Λ**, which are the eigenvector matrix and diagonal eigenvalues matrix, respectively.

The eigenvectors represent graph frequency modes and are used for GFT. The GFT of brain signal $\boldsymbol{x} \in \mathbb{R}^{N \times 155}$ is obtained as

$$\tilde{\boldsymbol{x}} = \boldsymbol{V}^T \boldsymbol{x}. \tag{2}$$

The inverse GFT (IGFT) of $\tilde{\mathbf{x}}$ is attained by

$$\boldsymbol{x} = \boldsymbol{V}\tilde{\boldsymbol{x}}. \tag{3}$$

Notably, the eigenvector corresponding to the larger eigenvalue shows more variance and can pass higher graph frequencies [48]. Higher frequency modes can transform brain signals of

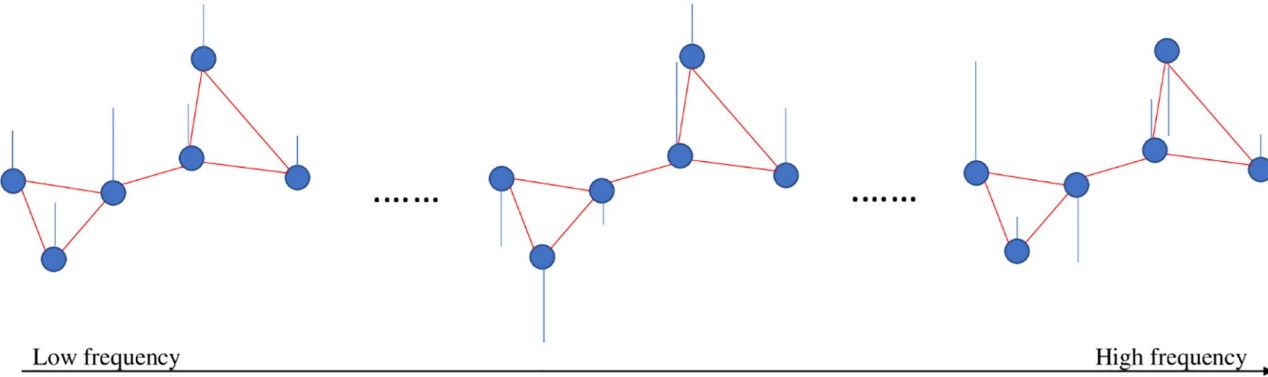

**Fig 2. Simple representation of frequency concept in graph domain.** In this domain, the signal changes across connected vertices define frequency levels (in the time domain, the signal changes across time points define frequency levels). Therefore, by moving from the lower frequency level to the higher frequency level of the graph, signal changes across connected vertices are increased. Blue circles and red and blue lines are vertices, edges, and signals, respectively.

higher variance to the graph frequency domain and inversely transform the higher frequency information from the frequency domain to the brain topological domain.

The graph signal can be filtered at the frequency domain and, then, got IGFT to have a graph filtered signal. The graph filtering process can be formulated as

$$x_F = VGV^Tx \tag{4}$$

where **G** is a diagonal filtering matrix. This study considered 1 for the diagonal elements corresponding to the desired frequency modes and 0 for the rest of the modes.

In this study, the first 33 and the last 33 frequency modes formed the LFB and HFB; the rest 34 modes formed the MFB. Using the "(4)", the rfMRI data was filtered to have data in graphs LFB, MFB, and HFB. Then, for each subject, the FC matrix was computed in the LFB, MFB, HFB, and FFB. It should be mentioned that the size of LFB, MFB, HFB, and FFB data matrices are equal to $N \times 155$ in this study. The N is the number of ROIs.

### Functional connectivity matrix

Recently, high-order FC (HOFC) methods have been used for obtaining the FC matrices [49, 50]. This study used the topographical profile similarity-based HOFC (tHOFC) [49]. In this method, the Pearson correlations of a given ROI with the rest of the ROIs are computed to be obtained the low-order FC (LOFC) profile of the given ROI. This computation is performed for each of the ROIs. Then, tHOFC is measured as the similarity of LOFC profiles between each pair of brain regions. The tHOFC can offer supplementary information to the conventional LOFC and introduce more differences between groups [49, 50].

To remove spurious connections and to obtain sparsely connected matrices, proportional thresholding was applied on FC matrices [51]. This thresholding provides an equal number of connections and connection density for all subjects of ASD and TC groups [51, 52]. Applying this thresholding preserves PT% of the strongest connections. In this study, the PT was changed from 10 to 50 with a step of 5. The PTs < 40 resulted in a fragmented graph for some subjects in the MFB and HFB. The PTs of 40 and 45 offered similar results. The PT = 40 was selected as the final threshold value to have more sparse matrices. The sparse connected matrices were used for scenarios of this study.

Connectivity matrices were calculated using the GraphVar software [53]. The rfMRI data of LFB, MFB, HFB, and FFB were separately loaded in this software. Then, the calculation of tHOFC was performed for the studied subjects. All 155 time points of ROIs were used to compute one tHOFC matrix for each subject. Thus, for calculating the studied metrics, we did not deal with 155 time points of ROIs. Instead, we dealt with one sparse connectivity matrix (sparse tHOFC matrix).

## Scenarios

The sparse tHOFC matrices were used to calculate the studied metrics. The global metrics of the graph were computed by GraphVar software [53]. Personal codes performed the analysis of triadic interactions. There was no thresholding on the triads. The thresholding was only applied to the tHOFC matrix for attaining a sparse matrix. Then, the triads were computed using sparse tHOFC matrices.

In the LFB, MFB, HFB, and FFB, the metrics were statistically compared between ASD and TC groups. Also, for a given group, the metrics were statistically compared between GFBs.

The scenarios of this study were implemented using data of 52 ASD patients and 44 TC subjects of NYU1 and SDSU datasets. The dimension of tHOFC connectivity matrices was 100*100 and 200*200 for the Schaefer atlas with 100 ROIs and 200 ROIs, respectively.

**Global metrics of graph.** The studied graph global metrics were assortativity, clustering coefficient, efficiency, radius, diameter, strength, SWN, and small-world propensity (SWP) [15].

The **assortativity** coefficient is a Pearson correlation between the degrees of all vertices on two opposite ends of a link (the degree of a vertex is the number of edges connected to it). This coefficient takes values between -1 and 1. The positive values show that vertices with similar degrees tend to connect. A graph's negative assortativity value states that vertices with larger degrees tend to connect to vertices with smaller degrees.

The **Clustering coefficient** is the fraction of triangles around a vertex and takes values between 0 and 1. The value 1 means connected vertices to a given vertex are also connected. Fewer connections in the neighborhood of a given vertex lead to less value of the clustering coefficient. This study reported the clustering coefficient value averaged over ROIs for each subject.

The **efficiency** is the average inverse shortest path length in the graph. In a weighted graph, the shortest path length between two vertices equals the minimum sum of edge weights between them. This metric measures the efficiency of information exchanging in the graph. The **efficiency** can be formulated as

$$\textbf{\textit{Efficiency}} = \left( \sum_{i \neq j} 1/l_{ij} \right) / M(M-1) \tag{5}$$

where $l_{ij}$ is the shortest path length between $i^{th}$ and $j^{th}$ ROIs, and $M$ represents the total number of ROIs.

Among all the maximum distances between a vertex and all other vertices, the radius/diameter is the minimum/maximum. The **strength** is the sum of weights of adjacent edges to the vertex. This study reported the strength value averaged over ROIs for each subject.

The **SWN**s are between lattice and random graphs, showing high clustering coefficients and low average shortest path lengths ($L_{avg}$). Such high and low properties make SWNs suitable for representing real-world networks such as brain networks. In the brain, the high clustering coefficient indicates functional specialization (processing information within each cluster of brain regions), and low $L_{avg}$ represents the efficient functional integration of brain sub-graphs [15, 41]. A formal analysis should determine which clustering coefficient is "high"

and which path length is "low" in a network. To address this problem, the normalized clustering coefficient and $L_{avg}$ are used to calculate SWN metric. Thus, the SWN metric ($\sigma$) is computed as

$$\sigma = \gamma/\lambda, \ \ \gamma = CC/CC_{rand}, \ \ \lambda = L_{avg}/L_{avg\_rand}, \tag{6}$$

where $CC_{rand}$ and $L_{avg\_rand}$ are computed in an ensemble of randomized surrogate networks. In a small-world network, a comparable path length and a higher clustering coefficient than a random network are expected ($\lambda \sim 1, \gamma > 1, \sigma > 1$) [15].

The definition of "(6)" has some limitations. Firstly, a network may have $\sigma > 1$ even when its $L_{avg}$ is much greater than $L_{avg}$ of a random graph. This sentence means that a small world network always has $\sigma > 1$, but a network with $\sigma > 1$ is not necessarily a small world. Secondly, the $\sigma$ is sensitive to the density of the network. The denser networks have smaller $\sigma$ even if they are generated from an identical small-world model [41]. To overcome these and other limitations, Muldoon and colleagues developed SWP by employing the $CC$ and $L_{avg}$ of both lattice and random networks [54]. The SWP measure $\phi$ is computed as

$$\phi = 1 - \sqrt{(\Delta C^2 + \Delta L^2)/2}, \tag{7}$$

where

$$\Delta C = (CC_{latt} - CC)/(CC_{latt} - CC_{rand}) \tag{8}$$

and

$$\Delta L = \left(L_{avg} - L_{avg\_rand}\right)/(L_{latt} - L_{rand}). \tag{9}$$

The authors set the $\Delta C$ and $\Delta L$ to 1/0 when they were larger/smaller than 1/0. By doing so, it is guaranteed that the $\phi$ is bounded in the range [0,1]. Networks with $0.4 < \phi \le 1$ are considered small-world [41].

**Triadic interactions and their metrics.** In this study, as with Moradimanesh and colleagues [24], four types of triads were analyzed in the structural graph's LFB, MFB, and HFB. These triads were strongly balanced $T_3$: ($+ + +$), weakly balanced $T_1$: ($+ - -$), strongly unbalanced $T_2$: ($+ + -$), and weakly unbalanced $T_0$: ($- - -$). Two metrics were used for comparing the triadic interactions between ASDs and TCs. The first one was the number of the triad, also called the frequency of triad ($|T_i|$, i = 0,1,2,3). The second one was the energy of the whole-brain network (Un). The Un is defined as

$$Un = -\sum_{i=0}^{i=3} \sum_{x<y<z} w_{xy}(T_i) w_{xz}(T_i) w_{yz}(T_i)/\Delta \tag{10}$$

where x, y, and z indicate the ROIs of triad $T_i$, $W_{xy}$ is the FC value, and $\Delta$ is the total number of triads of the brain.

In the brain network, it was shown that balanced/unbalanced triads were over-/under-presented [24]. This means that the frequencies of balanced/unbalanced triads of the real network were more/less than those of random networks. The surprise value ($S(T_i)$) is a metric providing positive/negative values for over-/under-presented triads [24]. This metric is defined as

$$S(T_i) = (T_i - E[T_i])/\sqrt{\Delta p_o(T_i)(1 - p_o(T_i))} \tag{11}$$

where $E[T_i] = p_o(T_i)\Delta$ is the expected number of triads $T_i$ and $\Delta$ is the total number of triads in the random network. The $p_o(T_i)$ is the ratio of the number of triad $T_i$ to the total number of triads in the random network. The random network had the same number of positive and

negative links as the real brain network and the signs of the links were randomly assigned to the existing links. In this study, the $S(T_i)$ was only used to provide information for the over-/under-presented behavior of triads in the graph's LFB, MFB, and HFB.

## Statistical analysis

The general linear model (GLM) was applied with age, diagnosis, and site variables as between covariate, between factor, and nuisance covariate, respectively. The diagnosis variable was set to 1 and 0 for ASD and TC subjects, respectively. The site variable was set to 1 and 0 for SDSU and NYU1 data, respectively. The nuisance covariate means that the site effect was regressed out from the dependent variable (studied metrics) before performing GLM. The model for analyzing the impact of the diagnosis variable is defined as

$$y = \beta_0 + \beta_{age} x_{age} + \beta_{diagnosis} x_{diagnosis} + \beta_{age\_diagnosis} x_{age} x_{diagnosis} + \epsilon \qquad (12)$$

where $\mathbf{y}$ is a vector containing ASD and TC measures from which the site effect was regressed. This model was performed using MATLAB software's command "stepwiseglm" with the constant model as the starting model, the interaction model as the upper model, and deviance as the criterion for adding or removing terms [10].

The GLM was for between-group comparisons. For a given group, the comparison between GFBs was performed using the Wilcoxon rank-sum test [55] as a nonparametric test was used for statistical comparison purposes.

The non-parametric permutation testing with 1000 repetitions was performed to correct all the *p-value* results. For between groups/GFBs comparisons, in each repetition, the measured value of each subject was randomly assigned to one group/GFB so that the final numbers of subjects of groups/GFBs were equal to the numbers of original groups/GFBs. Then, the difference of measure between two groups/GFBs was calculated. This process was repeated 1000 times to obtain a distribution of measure difference. The probability of the original between-group/between-GFB difference was calculated as its percentile position in this distribution.

## Results

In this section, the degree of freedom of all *t* values is 96, and the *p-values* are corrected.

### Results of graph metrics

The results of the statistical comparison are listed in Table 2. The statistically significant differences between ASDs and TCs were only seen in the MFB and HFB. The results of GFBs are shown in Fig 3.

For the **assortativity**, there was seen no significant difference between groups and between GFBs. This meant that both groups had similar resistance against failures in the main components of their network (i.e., their ROIs and edges) [16]. Such similarity was not sensitive to graph frequencies.

For the **clustering coefficient**, the ASDs took higher values than TCs in all GFBs, particularly in the MFB and HFB. However, the dominancy of ASDs was significant only inthe MFB ($t = 2.11$, $p = 0.031$). Both groups showed a similar pattern of clustering coefficient changes across GFBs i.e., decreased clustering coefficient by moving from LFB to MFB and increased clustering coefficient by moving from MFB to HFB. For TC group, the clustering coefficient values in the MFB and HFB were significantly lower than clustering coefficient value in the LFB ($z = -2.82$, $p = 0.001$ for MFB; $z = -1.81$, $p = 0.009$ for HFB).

**Table 2. Statistical comparison between ASDs and TCs for global metrics of the graph.** The *t* and *p* are statistical and corrected probability values, respectively.

|  |  | LFB | | MFB | | HFB | | FFB | |
|---|---|---|---|---|---|---|---|---|---|
|  |  | *t* | *p* | *t* | *p* | *t* | *p* | *t* | *p* |
| ASD vs TC | **Assortativity** | 0.27 | 0.85 | 0.48 | 0.56 | -0.25 | 0.9 | -0.03 | 0.9 |
|  | **Clustering coefficient** | 0.22 | 0.72 | 2.11 | **0.031** | 1.7 | 0.07 | -0.09 | 0.78 |
|  | **Efficiency** | -0.001 | 0.92 | 2.11 | **0.031** | 1.93 | **0.033** | 0.2 | 0.72 |
|  | **Radius** | -0.11 | 0.84 | -0.36 | 0.76 | -0.47 | 0.78 | -0.82 | 0.64 |
|  | **Diameter** | 0.73 | 0.43 | 0.73 | 0.31 | 1.3 | 0.37 | -1.69 | 0.21 |
|  | **Strength** | 0.06 | 0.86 | 1.94 | **0.037** | 1.93 | **0.034** | 0.1 | 0.8 |
|  | **SWP** | 0.75 | 0.45 | 0.68 | 0.18 | -1.98 | **0.027** | -0.02 | 0.78 |
|  | **SWN** | -1.38 | 0.27 | -1.15 | 0.12 | -1.38 | 0.1 | 0.39 | 0.41 |

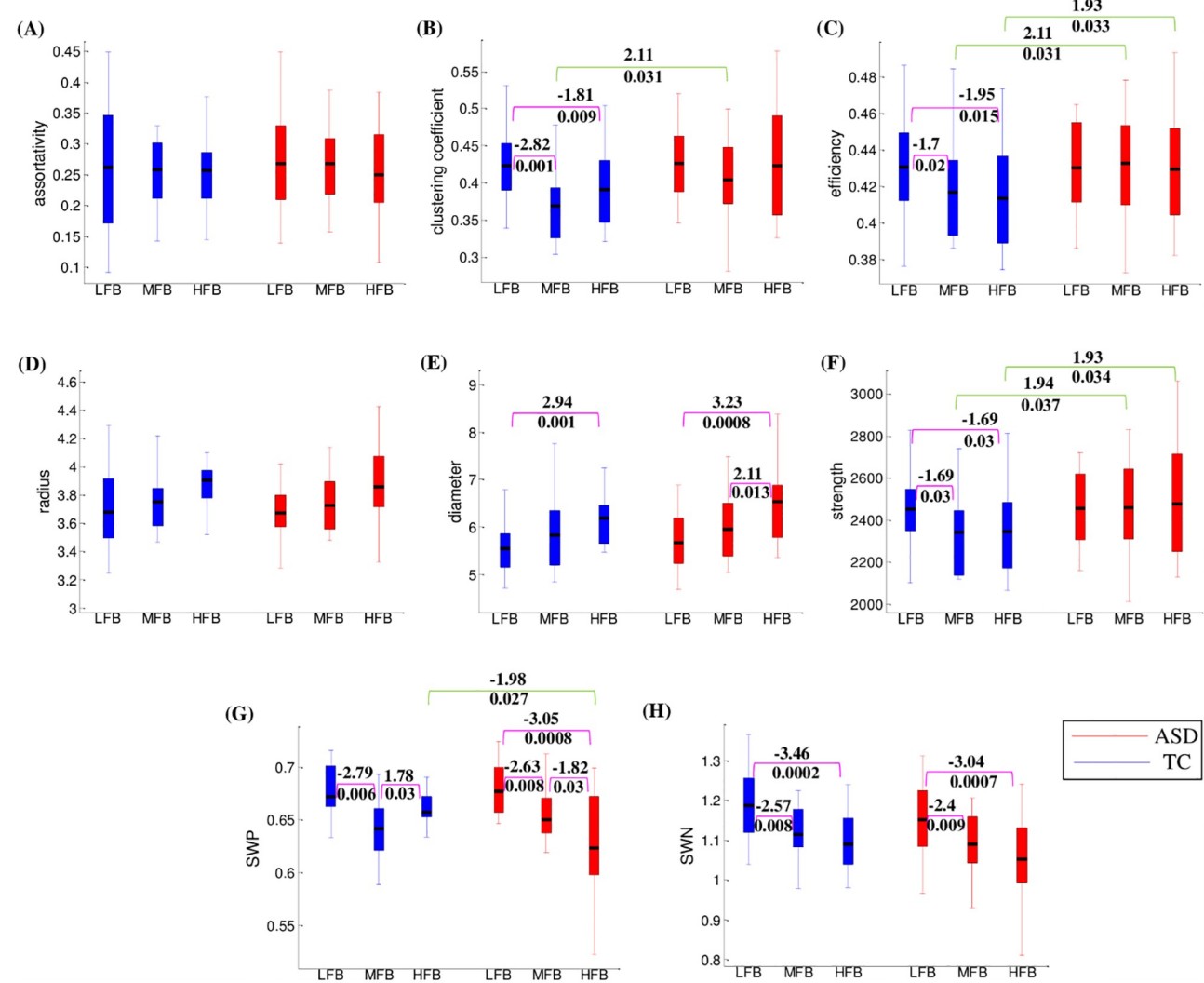

**Fig 3. Comparing the graph global metrics between groups and between graph frequency bands.** The results are for **(A)** Assortativity, **(B)** Clustering coefficient, **(C)** Efficiency, **(D)** Radius, **(E)** Diameter, **(F)** Strength, **(G)** SWP, and **(H)** SWN metrics. The green and pink color lines show the results between groups and frequency bands, respectively. The values above and under the lines are statistical t and corresponding corrected p values, respectively. The values mentioned above are only provided for significant results ($p < 0.05$).

For the **efficiency**, the ASD group had similar behavior across the GFBs. In contrast, the efficiency of the TC group was decreased by increasing the graph frequencies. As a result, significant differences between ASDs and TCs were seen in the MFB ($t = 2.11$, $p = 0.031$) and HFB ($t = 1.93$, $p = 0.033$). Also, there were significant differences between MFB and LFB ($z = -1.7$, $p = 0.02$) and between HFB and LFB ($z = -1.95$, $p = 0.015$) for TC group.

For the **radius** metric, there were no significant results. The common pattern for the two groups was increasing the radius by increasing the graph frequencies.

For the **diameter** metric, similar behavior to the radius metric was seen. However, the differences between GFBs reached a significant level. The diameter of HFB was significantly larger than that of LFB for both groups ($z = 3.23$, $p = 0.0008$ for ASDs; $z = 2.94$, $p = 0.001$ for TCs). For ASDs, another significant result was found between HFB and MFB ($z = 2.11$, $p = 0.013$).

For the **strength**, ASDs had significantly larger values than TCs in the MFB ($t = 1.94$, $p = 0.037$) and HFB ($t = 1.93$, $p = 0.034$). The strength values of the ASD group didn't show changes across GFBs. The strength values of the TC group were similar to each other in the MFB and HFB and were significantly lower than the LFB's ($z = -1.69$, $p = 0.03$ for both MFB and HFB).

SWP and SWN are metrics for investigating small-world properties of complex networks. However, the SWP is the updated and modified one [15, 41]. The between groups pattern, which was common in both SWP and SWN, was the lower small-world property of ASDs than TCs in the HFB. This pattern reached to the significant level when using SWP metric ($t = -1.98$, $p = 0.027$). For the ASD group, the GFBs pattern, which was common in both SWP and SWN, was the reduction of small-world property by increasing graph frequencies. For SWP, significant differences were seen between MFB and LFB ($z = -2.63$, $p = 0.008$), between HFB and MFB ($z = -1.82$, $p = 0.03$), and between HFB and LFB ($z = -3.05$, $p = 0.0008$). For SWN, significant differences were only seen between MFB and LFB ($z = -2.4$, $p = 0.009$) and between HFB and LFB ($z = -3.04$, $p = 0.0007$). For the TC group, the common pattern was the lower small-world property in the MFB and HFB compared to LFB. For SWP, there were significant difference between MFB and LFB ($z = -2.79$, $p = 0.006$) and between HFB and MFB ($z = 1.78$, $p = 0.03$). For SWN, significant results were seen between MFB and LFB ($z = -2.57$, $p = 0.008$) and between HFB and LFB ($z = -3.46$, $p = 0.0002$).

## Results of triadic interactions

For ASD and TC groups, the frequency of triads ($|T_i|$), surprise value S, and the ratio of $p/p_0$ are listed in Table 3. In FFB and all GFBs and for both groups, the unbalanced/balanced triads ($T_o$ and $T_2$)/($T_1$ and $T_3$) were under-presented/over-presented. The under-presented/over-presented shows itself by ($p/p_0 < 1$ and $S < 0$)/($p/p_0 > 1$ and $S > 0$). The $p$ ($T_i$) is the ratio of the number of triad $T_i$ to the total number of triads in the original network.

The information in this paragraph is for both ASD and TC groups. In FFB and all GFBs, the number of balanced triads ($|T_1|$ and $|T_3|$) was more than that of unbalanced triads ($|T_0|$ and $|T_2|$). In FFB and all GFBs, the $|T_0|$ was the minimum number of triads. The maximum number of triads in the LFB was devoted to the $T_3$. However, in the higher graph frequencies (MFB and HFB) and FFB, the $|T_1|$ was the maximum. It should be mentioned that the ratios of $|T_1|/|T_3|$ in the MFB, HFB, and FFB were much bigger than the ratio of $|T_3|/|T_1|$ in the LFB. On average, the total number of triads in the (LFB, MFB, HFB, FFB) were (86648,86002,86875, 88310) and (86043,85333,85950,88598) for ASDs and TCs, respectively.

**Table 3. The metrics |T$_i$|, S, and the ratio of p/p$_0$ averaged over subjects.**

| | | ASD | | | | TC | | | |
|---|---|---|---|---|---|---|---|---|---|
| | | T$_0$ | T$_1$ | T$_2$ | T$_3$ | T$_0$ | T$_1$ | T$_2$ | T$_3$ |
| LFB | \|T$_i$\| | 544 | 32912 | 10210 | 42982 | 526 | 29463 | 10982 | 45072 |
| | p/p$_0$ (T$_i$) | 0.1804 | 1.963 | 0.292 | 1.3578 | 0.207 | 1.954 | 0.317 | 1.339 |
| | S (T$_i$) | -42.03 | 142.4 | -173.35 | 85.28 | -36.66 | 129.98 | -165.35 | 84.37 |
| MFB | \|T$_i$\| | 3135 | 56434 | 8826 | 17607 | 3781 | 53916 | 10770 | 16866 |
| | p/p$_0$ (T$_i$) | 0.28 | 1.72 | 0.28 | 1.72 | 0.34 | 1.66 | 0.34 | 1.65 |
| | S (T$_i$) | -82.4 | 165.6 | -161.2 | 77.9 | -74.791 | 150.77 | -146.83 | 70.541 |
| HFB | \|T$_i$\| | 2766 | 56958 | 8281 | 18870 | 3255 | 55276 | 9606 | 17813 |
| | p/p$_0$ (T$_i$) | 0.26 | 1.74 | 0.26 | 1.75 | 0.298 | 1.7 | 0.3 | 1.69 |
| | S (T$_i$) | -83.332 | 169.425 | -169.23 | 83.38 | -78.9 | 160.4 | -157.9 | 76.23 |
| FFB | \|T$_i$\| | 3453 | 58153 | 8670 | 18034 | 3298 | 58518 | 8465 | 18317 |
| | p/p$_0$ (T$_i$) | 0.29 | 1.71 | 0.27 | 1.75 | 0.28 | 1.72 | 0.26 | 1.76 |
| | S (T$_i$) | -82.47 | 167.52 | -164.8 | 81.02 | -83.9 | 169.16 | -166.83 | 82.76 |

The results of the statistical comparison are listed in Table 4. The statistically significant differences between ASDs and TCs were only seen in the MFB and HFB. The results of GFBs are shown in Fig 4. The Log-Log distribution of Un at LFB, MFB, and HFB are plotted in S3 Fig in S1 File.

For both groups, by moving from LFB to higher graph frequencies, significant high-level changes were seen for |T$_0$|, |T$_1$|, and |T$_3$|. The greatest changes were for |T$_0$| and |T$_3$|. The earlier one increased by a factor of 5–6, and the latter decreased by a factor of 3. The |T$_1$| increased by a factor of close to 2. The significance between-groups differences were found in the MFB (for |T$_0$|, |T$_1$|, and |T$_2$|) and HFB (for |T$_3$|). The number of unbalanced triads T$_0$ and T$_2$ were significantly lower for ASDs compared to TCs in the MFB ($t = -2.03$, $p = 0.01$ for T$_0$; $t = -2.12$, $p = 0.01$ for T$_2$). In contrast, the |T$_1$| and |T$_3$| were significantly higher for ASDs compared to TCs in the MFB ($t = 1.89$, $p = 0.02$) and HFB ($t = 2.02$, $p = 0.01$), respectively. The general pattern in the MFB and HFB was a higher/lower number of balanced/unbalanced triads for ASDs compared to TCs.

The **energies** (Un) of the two groups were negative and approximately equal in the LFB. For TCs, the energies in the MFB and HFB were more than the energy in the LFB. In contrast, increasing the graph frequencies was accompanied by decreasing the energy level for ASDs. Consequently, significant differences were seen between energies of ASD and TC groups in the MFB ($t = -1.69$, $p = 0.02$) and HFB ($t = -1.69$, $p = 0.022$).

**Table 4. Statistical comparison between ASDs and TCs for metrics of triadic interactions.** The $t$ and $p$ are statistical and corrected probability values, respectively.

| | | LFB | | MFB | | HFB | | FFB | |
|---|---|---|---|---|---|---|---|---|---|
| | | t | p | t | p | t | p | t | p |
| ASD vs TC | T$_0$ | 0.1 | 0.61 | -2.032 | **0.01** | -1.44 | 0.19 | 0.39 | 0.81 |
| | T$_1$ | 0.86 | 0.15 | 1.89 | **0.02** | 1.11 | 0.09 | -0.19 | 0.69 |
| | T$_2$ | -0.72 | 0.19 | -2.125 | **0.01** | -1.28 | 0.09 | 0.22 | 0.72 |
| | T$_3$ | -0.51 | 0.16 | 1.64 | 0.14 | 2.024 | **0.01** | -0.39 | 0.5 |
| | Un | -0.02 | 0.8 | -1.69 | **0.02** | -1.69 | **0.022** | 0.28 | 0.83 |

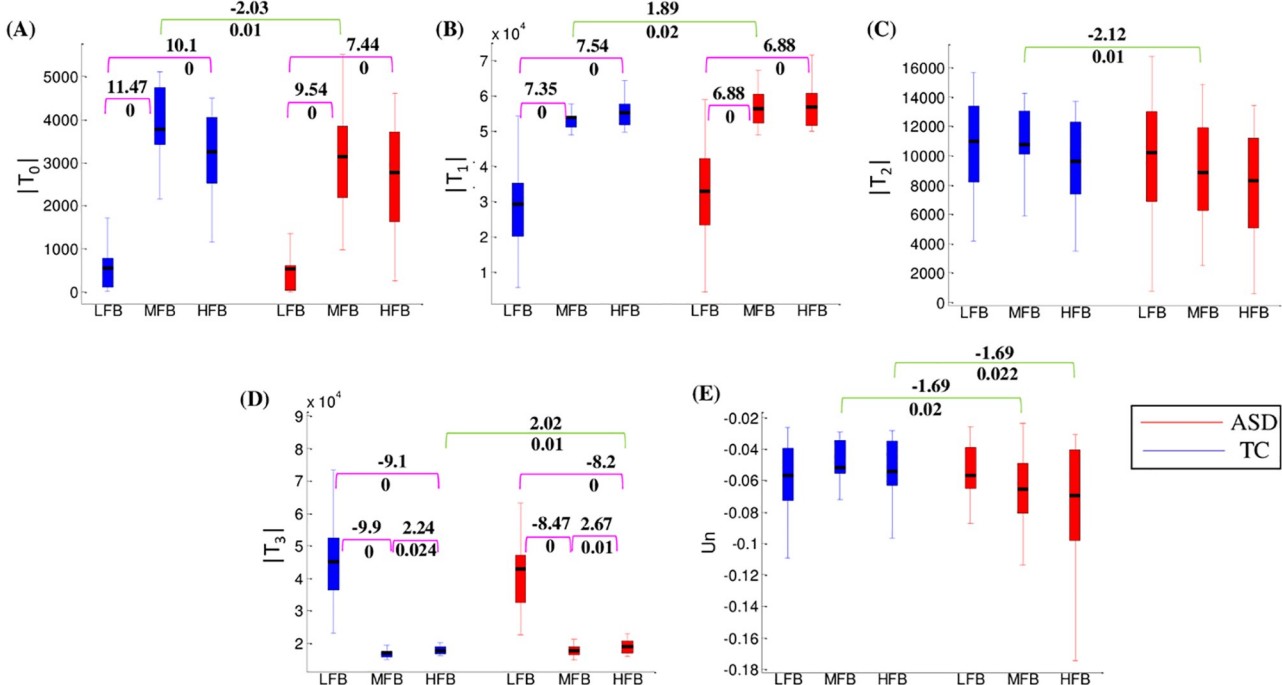

**Fig 4. Comparing the triadic interaction metrics between groups and between graph frequency bands.** The results are for **(A)** $|T_0|$, **(B)** $|T_1|$, **(C)** $|T_2|$, **(D)** $|T_3|$, and **(E)** Un metrics. The green and pink color lines show the results between groups and frequency bands, respectively. The values above and under the lines are statistical t and corresponding corrected p values, respectively. The values mentioned above are only provided for significant results ($p < 0.05$).

## Discussion

In this paper, unfiltered and topologically filtered rfMRI data of ASDs and TCs were studied to reveal the value of structural graph frequency bands. The studied features were global metrics of functional graphs and metrics of functional triadic interactions.

The results of Tables 2 and 4 demonstrated that frequency bands of the structural graph could introduce features discriminating between ASDs and TCs which were not discriminative in the FFB data. These results show the promising perspective of analyzing the functional data in the structural graph frequency bands. It should be noted that some studies have found discriminative features of MFB and HFB in the unfiltered rfMRI data [17, 18, 20, 24, 37, 38] (the findings of these studies are referred to in the following paragraphs of the discussion). Conditions such as using both males and females, different age ranges, different FCs, and heterogeneity of ASD can be the possible reasons. However, it should be noted that the results of LFB, MFB, HFB, and FFB were obtained under the same conditions in this study. This means that GFBs in comparison to FFB can provide significant features under the same conditions. The clustering coefficient, efficiency, and strength were found as discriminative features, consequently providing more information. Only one electroencephalography (EEG) study reported clustering coefficient, efficiency, and strength as discriminative features [37]. Their findings were distributed in the theta, beta, low, and high gamma bands. In none of the previous rfMRI studies, all of these metrics were not discriminative between ASDs and TCs, simultaneously. This implies that frequency bands of the structural graph can reveal more intrinsic like as EEG study in the theta, delta, alpha, beta, and gamma bands can provide more band-specific information. One of the main findings of this study was that significant

differences between ASDs and TCs existed in the MFB and HFB. In the following, these differences are discussed.

The average clustering coefficient of a network is a direct metric for measuring the local specialization of the network [12]. In the MFB, the clustering coefficient of ASDs was significantly larger than TCs. Thus, the results of the clustering coefficient in the MFB may show a decrease in brain functional specialization in ASDs due to an abundance of functional connections (increased clusters of local connections in ASDs). Similar results have also been reported in the DTI study [56] and the electroencephalography (EEG) study in the theta, beta, and low gamma bands [37]. In the MFB and HFB, the connection strength of ASDs was significantly larger than TCs. This result may show the over-connectivity in ASDs compared to TCs in the MFB and HFB. The over-connectivity has been reported in many ASD studies [17, 18, 37, 38]. Increased strength and local connectedness in ASDs may suggest impaired network refinement in high-frequency bands of the structural graph [37]. The efficiency results show that the global integration of ASDs is more than TCs in the MFB and HFB [20, 37, 56]. Increased clustering coefficient and efficiency indicate increased "cliquishness" properties of functional networks of ASDs in the MFB and HFB, supporting the idea of a disrupted balance between global integration and local specialization in ASDs. Similar results have also been seen in theta, beta, low, and high gamma bands in an EEG study [37]. ASD and TC groups had small-world properties ($\sigma > 1$, $\phi > 0.4$). However, the ASDs took lower $\sigma$ and significantly lower $\phi$ than TCs in the HFB. These results show the reduction of optimal balance between network segregation and integration in ASDs.

The clustering coefficient and strength results indicated the local over-connectivity and greater local efficiency in ASDs. The efficiency and strength results suggest a pattern of global overconnectivity. All of these indicate the disruption of the balance between network segregation (revealed by clustering coefficient and strength) and integration (revealed by efficiency and strength) in ASDs. This conclusion is in line with the results of SWN and SWP. Several studies reported disruption between segregation and integration [37, 57–59].

One of the results of this study was local functional over-connectivity in ASDs. There are some pathophysiological evidence and findings that support this result. One of the findings is the imbalance in excitatory/inhibitory neural activity [60]. Diminished GABAergic function is hypothesized to disrupt the excitatory/inhibitory balance at the neuronal level, leading to mostly over-connectivity of ROIs (networks) in ASD [61]. Another pathophysiological finding is that people with autism do not undergo normal pruning during childhood and adolescence [62]. A synapse allows for neural communication between cells. When there are too many synaptic connections, the brain goes through a process of cutting down, known as synaptic pruning (cutting out the non-functional, unnecessary neurons to increase the power of the working ones). Studies in neural density have found a higher number of neurons among autistic individuals due to a slowdown in a normal brain "pruning" process during development [62–64]. It has been proposed that an excess of neurons causes local functional over-connectivity [65, 66]. Thus, our findings may reflect an immature connectivity pattern in ASDs.

In this study, Heider's balance theory was seen for ASD and TC groups in all GFBs [67]. This meant that all balanced/unbalanced triads were over-presented/under-presented in the brains of ASDs and TCs in all GFBs (Table 3). However, as has been reported by Moradimanesh and colleagues [24] and some studies of social networks [68, 69], $T_0$ may show over-presented behavior in some cases. Thus, this issue was separately investigated on ASDs and TCs in the LFB, MFB, and HFB. Only 3.84% of ASD subjects showed over-presented $T_0$ in the LFB. However, this percent value is small, and the dominance behavior for $T_0$ is under-presentation in the ASDs.

The unbalance triads inject energy into the brain and excite it to change its state, resulting in brain dynamism [24, 67]. Thus, the unbalanced triads play a crucial role in the brain's dynamism, although they are under-presented. In this study, the numbers of unbalanced triads $T_0$ and $T_2$ were significantly lower in ASDs compared to TCs in the MFB. Also, the correlations of $|T_0|$ and $|T_2|$ with Autism Diagnostic Interview restricted and repetitive behavior (ADIRRB) score were computed. Negative associations with ADIRRB were seen for both of $|T_0|$ ($r (46) = -0.37; p = 0.01$) and $|T_2|$ ($r (46) = -0.31; p = 0.036$) (the scores of 46 subjects were available). As a result, it can be said that the lack of enough numbers of unbalanced triads in the MFB leads to difficulty in dynamic switching of the brain, which in turn may be a reason for increased repetitive and restricted behaviors in ASDs.

The role of balanced triads from the perspective of structural balance theory is to provide connected modularity. This means balanced triads provide connected clusters of ROIs for the brain in its stable state [24]. Thus, the role of balanced triads is important in balancing the functional specialization and integration of brain networks. In this study, the numbers of balanced triads $T_1$ and $T_3$ were significantly larger for ASDs compared to TCs in the MFB and HFB, respectively. Indeed, these differences result in an imbalance of brain network integration and specialization. These findings are in line with the clustering coefficient results and global efficiency in which decreased local specialization and increased global integration were seen in the MFB and HFB.

For ASDs, having lower numbers of unbalanced triads and higher numbers of balanced triads can be reasons for having lower energies compared to TCs in the MFB and HFB. This means that lower energies of the ASD group can show both or each of the lower dynamism and unbalanced functional integration and specialization of the brain. To clarify, the correlations of Un with ADIRRB, clustering coefficient, and global efficiency were computed in the MFB and HFB. The correlations with ADIRRB offered ($r (46) = -0.16, p = 0.29$) and ($r (46) = -0.07, p = 0.64$) in the MFB and HFB, respectively. The correlations with clustering coefficient and global efficiency were significant in both MFB and HFB (*all $r < -0.9$, all $p < 1e-18$*). As a result, it can be said that lower energies of ASDs are due to having more balanced triads. From another point of view, lower energies of ASDs indicate the disruption of the optimum balance between functional integration and local specialization in the MFB and HFB.

Most of the significant results of this paper were found in the MFB and HFB. Accordingly, the belief that only the first several graph frequency modes contain the most critical information, just as the classical Fourier frequency [70], cannot be valid [48], at least for comparison analysis of rfMRI data of ASD subjects and TCs. From another perspective, given the frequency illustrated in Fig 2, it is evident that the rfMRI data cannot be similar (with the same sign and very low variability) for all connected ROIs. In other words, the brain functional data also exists in the MFB and HFB of the structural graph.

The scenarios of this paper were also implemented using Schaefer atlas with 200 ROIs (FA200) to investigate the reproducibility of results with different numbers of ROIs. For all metrics, the results of FA200 were similar to those of FA100 and the significant results were repeated. The results regarding 200 ROIs are represented in S.1 of the S1 File for interested readers. By reviewing the results of S1 and S3 Tables in the S1 File, Tables 2 and 4, it was seen that the signs of statistical values (*t*) were only preserved in the MFB and HFB while changing the number of ROIs. By reviewing the results of the S2 Table in S1 File and Table 2, it was seen that the ratio of $|T_3|$ to $|T_1|$ was larger than one for FA200 whereas this ratio was smaller than one for FA100. Also, the ratios of $|T_3|$, $|T_2|$, and $|T_1|$ to $|T_0|$ were much larger for FA200 than FA100. These results may be interpreted as the higher reproducibility of results in the MFB and HFB compared to LFB and FFB.

All the graph metrics (except SWN) were computed together in the GraphVar software. The computation time of these processes was about 2 minutes. The mentioned processes were repeated for SWN with a computation time of approximately 450 minutes. The computation time of all triad metrics (together) was about 1 minute. These times are times needed to compute metrics for all studied subjects. All the computation was performed using a PC Core i7-8700k @ 3.7GHz.

Community detection analysis was also performed using the Louvain method in the GraphVar software [53, 71]. The more detailed results are given in section S.3 of the S1 File. In each GFB, three modules were found for ASDs and TCs (S1 Fig in S1 File). There were no significant differences between ASDs and TCs when investigating the maximum modularity ($Q$), normalized mutual information ($MIn$), and normalized variation of information ($VIn$) metrics (these metrics are measures of the modular organization of brain networks (more explanations are in S.2 of S1 File)). However, the $VIn$ was much more than $MIn$ in the HFB and, particularly, in the MFB. This meant that modules of ASDs and TCs had less common information and more variation from each other in the MFB and HFB. The modularity overlap information confirmed the $VIn$ and $MIn$ results (S4 and S5 Tables in S1 File). These results were consistent with our main finding that the significant differences between ASDs and TCs existed in the MFB and HFB when analyzing the global metrics of the graph and triadic interaction metrics.

The ROIs were investigated for translational and local connectivity hub roles in the community analysis. Transitional ROIs facilitate functional integration between modules. Hub ROIs of a module have many connections with other ROIs of that module i.e., they are the core components of that module. In the LFB, the translational roles of the right hemisphere somatomotor 7 and 8 were significantly more for TCs than ASDs (transitional ROIs facilitate functional integration between modules). In the MFB and HFB, there were no ROIs with significant differences between ASDs and TCs regarding the translational role. For the hub role, the left hemisphere lateral prefrontal cortex (salience/ventral attention network) in the LFB and the right hemisphere prefrontal cortex 6 (ROI of default mode network) and visual 1 in the MFB were for ASDs. For TCs, the left hemisphere precuneus posterior cingulate cortex 1 (ROI of default mode network) in the MFB and right hemisphere visual 7 in the HFB were the local hub.

In this study, an equal number of frequency modes were considered to make LFB, MFB, and HFB. This division was token because it was the most straightforward way and the first thing that comes to mind. Such division offered results for graph metrics, triadic interaction metrics, and community structure analysis which were in line with each other and confirmed each other. Thus, such in-line results provided reliability for us. Also, the results of FA200 confirmed that equal division is trustable. However, sensitivity analysis may be helpful to see how stable the results are to the choice of the range chosen for assigning the frequency modes. For this analysis, the between-group results of triadic interaction metrics were studied. The frequency modes of LFB, MFB, and HFB were considered as $[v_1, v_2, \ldots, v_L]$, $[v_{ML}, v_2, \ldots, v_{MH}]$, and $[v_H, v_{H+1}, \ldots, v_{100}]$, respectively. In this study, the values of $L$, $ML$, $MH$, and $H$ were 33, 34, 67, and 68, respectively. To investigate the sensitivity, the values of $L$, $ML$, $MH$, and $H$ were decreased and increased by 1, 2, 3, 4, and 5. Thus, the $L$ of $v_L$ was changed from 28 to 38 and no significant alternations were seen in the results of LFB, i.e., changing from not significant to significant didn't happen. Also, by decreasing and increasing $ML$, $MH$, and $H$ there were no meaningful alternations in terms of changing significant results to not significant and vice versa. The behavior in LFB did not change even for $L = 20$ *or* 40. However, increasing the $H$ by 10 led to changing of Un and $|T_3|$ results from significant to not significant ($t = -1.09$, $p = 0.16$; $t = 1.6$, $p = 0.075$). Decreased $H$ by 10 led to no differences in significant results. Also, decreased and increased $ML$ and $MH$ by 10 led to no differences in significant results. These

results show the less sensitivity of our findings to the choice of the range chosen for assigning the frequency modes.

Although the ExploreDTI software was used herein, given its well-established reputation, fiber tractography generally has limitations in terms of false positives (and negatives). For insurance, the tractogram of each subject was filtered by linear fascicle evaluation (LiFE) procedure [72]. Then, the SC matrix of the filtered tractogram was computed as explained in subsection 2.2.2. For all subjects and atlases, the Pearson correlation analysis between original and filtered SC matrices showed no significance difference (all r > 0.76, all p << 0.0001). The general pattern of scenario results didn't show any changes, i.e., significant/non-significant results stayed significant/non-significant. Only some differences were seen in statistical *t* values and corresponding *p-values*. Thus, original and corrected SC matrices confirmed each other and were not discrepant each other.

## Limitations and future directions

In this paper, only male subjects with a broad age range (5.32–26.6 years) were studied. In the future, the data of both males and females with a narrower age range can be used to provide results for both genders. Overall, the enormous heterogeneity of ASD depends not only on age and gender but also on intellectual ability, genetic factors, and environmental risk factors [73]. Hence, if future studies consider these affective factors, they will bring more consistent data and repeatable results and improve understanding of the neurobiological mechanisms of ASD [8].

In this study, the underlying graph of GSP was attained using the SC of DTI data and the graph signal was rfMRI data. However, employing GSP using only one modality may be more convenient. To do this, one can use the FC of rfMRI as the underlying graph of GSP and investigate the reproducibility of the results of this study. However, as explained in the introduction section, it is convenient to have a stable topology. Hence, it is necessary to sparse FC matrix and preserves only the strongest connections. The stability of stronger functional connections is more than weaker ones [10, 74]. Such investigation and finding the proper FC matrix can be a topic for future work. Using a sparse FC matrix reduces sensitivity to inter-individual noise differences and global group connectivity differences (e.g., predominant under-connectivity or over-connectivity in ASDs) [19].

In this study, the graph frequencies were divided into three independent bands. For future work, one can divide graph frequencies into more GFBs. This may offer more discriminative features between ASDs and TCs.

The ASD researches dealing with global metrics of the graph are very limited. One reason might be that the researchers have not found these metrics as significant discriminative features. However, this study demonstrated that using GSP could make these metrics such discriminative features when using them in the MFB and HFB. For future work, global metrics of graph and triadic interaction metrics in the MFB and HFB can be used to classify ASDs and TCs. More investigation of these features using machine learning techniques and data from much more subjects may lead to introducing some of the studied metrics as biomarkers.

This study analyzed the connectivity matrices of rfMRI data through ordinary graph metrics. In an ordinary graph, an edge connects exactly two vertices. For future work, the hyper graph approaches can be used to compare filtered and unfiltered rfMRI data and to compare ordinary graphs versus hyper graphs in different frequency bands of the structural graph. A hypergraph is a generalization of a graph in which an edge can join any number of vertices [75].

For ASD and TC groups, the $|T_1|$ showed a significant increase in the MFB and HFB compared to LFB, whereas the $|T_3|$ showed a significant decrease in the MFB and HFB compared to LFB. The weakly balanced triads compared to strongly balanced ones are more prone to become unbalanced triads. As stated in the fourth paragraph of the discussion, the unbalanced triads are responsible for brain dynamism. Therefore, the resources for being more dynamic are more in the MFB and HFB. It might be guessed that having such high resources shows the brain's readiness for change and adaptation over time (age and neuroplasticity [76]). Investigation about this guess can be a topic for future work.

The community structure analysis revealed three modules (sub-networks) for ASDs and TCs in each studied GFBs. The studied metrics can be employed for future work to compare ASDs with TCs at these sub-networks.

## Conclusion

rfMRI data filtered by topological filter in contrast to unfiltered rfMRI data revealed significant differences between ASDs and TCs. The main finding was that significant differences between ASDs and TCs existed in the structural graph's MFB and HFB when analyzing the functional graph's global metrics and the brain's triadic interaction. ASDs in comparison to TCs had significant higher clustering coefficient in the MFB, higher efficiencies in the MFB and HFB, higher strengths in the MFB and HFB, lower SWP in the HFB, lower $|T_0|$ in the MFB, higher $|T_1|$ in the MFB, lower $|T_2|$ in the MFB, higher $|T_3|$ in the HFB, and lower energies in the MFB and HFB. These results imply that functional data passed through structural filters can introduce discriminative features which were not discriminatory in unfiltered functional data. Therefore, employing structural filters may provide a new avenue for extracting features that can be candidate biomarkers for ASD.

## Supporting information

**S1 File. Supplementary material to the manuscript.**
(DOCX)

## Acknowledgments

We would like to thank the autism brain imaging data exchange (ABIDE) for generously sharing the data.

## Author Contributions

**Conceptualization:** Alireza Talesh Jafadideh.

**Formal analysis:** Alireza Talesh Jafadideh.

**Investigation:** Alireza Talesh Jafadideh.

**Methodology:** Alireza Talesh Jafadideh.

**Project administration:** Babak Mohammadzadeh Asl.

**Software:** Alireza Talesh Jafadideh.

**Supervision:** Babak Mohammadzadeh Asl.

**Validation:** Alireza Talesh Jafadideh.

**Visualization:** Alireza Talesh Jafadideh.

**Writing – original draft:** Alireza Talesh Jafadideh.

**Writing – review & editing:** Alireza Talesh Jafadideh.

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
