## [Decision Letter · Decision Letter 0]

8 Aug 2022

PONE-D-22-18416Middle and High Frequency Bands of Structural Graph Provide Significant Functional Features for Autism Spectrum DisorderPLOS ONE

Dear Dr. Mohammadzadeh Asl,

Thank you for submitting your manuscript to PLOS ONE. After careful consideration, we feel that it has merit but does not fully meet PLOS ONE’s publication criteria as it currently stands. Therefore, we invite you to submit a revised version of the manuscript that addresses the points raised during the review process.

We look forward to receiving your revised manuscript.

Kind regards,

Moo K. Chung

Academic Editor

PLOS ONE

Journal Requirements:

Reviewers' comments:

Reviewer's Responses to Questions

**Comments to the Author**

1. Is the manuscript technically sound, and do the data support the conclusions?

Reviewer #1: Yes

Reviewer #2: Partly

Reviewer #3: Partly

2. Has the statistical analysis been performed appropriately and rigorously? 

Reviewer #1: Yes

Reviewer #2: No

Reviewer #3: I Don't Know

3. Have the authors made all data underlying the findings in their manuscript fully available?

Reviewer #1: Yes

Reviewer #2: Yes

Reviewer #3: Yes

4. Is the manuscript presented in an intelligible fashion and written in standard English?

Reviewer #1: Yes

Reviewer #2: Yes

Reviewer #3: Yes

5. Review Comments to the Author

Reviewer #1: Report on manuscript

Middle and High Frequency Bands of Structural Graph Provide Significant Functional Features for Autism Spectrum Disorder

by

Alireza Talesh Jafadideh , Babak Mohammadzadeh Asl

• Overview of manuscript

The authors investigate the possibility of using the topological filters to derive data with discriminative features between ASDs and TCs. They perform diffusion tensor imaging (DTI) to compute SC matrix. They obtain the topological filters using GFT tool and SC matrix. Using FC matrices, they derive the graph global metrics and triadic interaction metrics. They apply the topological filters on rfMRI data to obtain data in three independent frequency bands of structural graph. Some results are presented to compare the triadic interaction metrics between groups and between graph frequency bands.

• Comments on text

1. New contribution

GFBs in comparison to FFB can provide significant features under the same conditions.

The frequency bands of structural graph can reveal more intrinsic differences existing in functional data of ASDs and TCs, consequently, providing more information.

The significant differences between ASDs and TCs existed in the MFB and HFB.

2. English

The English in this paper is good.

Comments

The paper has sufficiently presented the research problem description, introduction, methods / techniques, results & discussion, concluding remarks and references.

However, the authors are pleased to consider the following suggestions to improve the quality of the article.

(a) I suggest the title be rephrased to be shorter but at the same time to capture the content and its significance of this research.

(b) “The general linear model (GLM) was applied with age, diagnosis, and site variables as between covariate” Did the others try other statistical modeling which is more accurate in most cases such as generalized linear mixed models (GLMM)? The authors need to provide more evidence on this model selection.

(c) The authors mentioned “The non-parametric permutation testing with 400 repetitions.” Could they verify how they obtained this number? Based on which analysis they concluded that “ This process was repeated 400 times to obtain a distribution of measure difference”.

(d) The authors need to provide more details on the “statistical comparison” test they have used. The same for global efficiency.

(e) The authors should take into account briefly answering this question in the conclusion part “What is the implication of your results in the related application area?”

(f) The authors may need to have more discussion on the connection between “abnormal connectivity in the ASD brains” as they explained in other studies in the introduction part and their results.

(g) For “Limitations and future directions”, the authors also can think about other factors since “Individuals with ASD show enormous heterogeneity depending on age, gender, intellectual ability, genetic factor, and environmental risk factor (Lenroot RK et al., 2013).” Since “Studies regarding these affective factors will bring more consistent data and improve understanding of neurobiological mechanisms of ASD (Sungji Ha et al., 2015).”

• Recommendation

After all the revisions, I can recommend acceptance of this paper.

Reviewer #2: This paper is overall coherent. The introduction provides reasonable background from the literature on the analytical pipeline used in the study. However, methodological components needs more justification rather than stating what they are and so and so paper used them. For instance, there are quite a few complex network measures (Rubinov and Sporns, 2010), or global metric of graph as the paper refers to them, that one can use. Why were assortativity, clustering coefficient, efficiency, radius, diameter, strength, SWN, and

small-world propensity (SWP) the only ones studied? Triadic interactions is an interesting piece of the study. But why use the approach of Moradimanesh et al. and not other hyper graph approaches? Statistical analysis consists of GLM and t-test, both of which rely on stringent distributional assumptions. How do we know that these assumptions are not violated by the complex neuroimaging data used in the study?

Reviewer #3: In this article, the authors investigate Can filtering the functional data through the frequency bands of structural graph provide data with valuable features for Austism Spectrum Disorder? The goal is to see if there are any interesting discriminative features which are otherwise not present in an unfiltered data in the ASD and Typical Controls (TC). The study uses standard graph metrics and a metric based on functional triadic interactions and the results show

differences in the ASD and TC using different frequency bands. This study on Structural filtering of functional data for feature extraction is interesting to read. The results indicate the usefulness of structural graph frequency bands to evaluate functional data allows to uncover features which are hitherto hidden is an important finding from this study.

Please see the attached comments

6. PLOS authors have the option to publish the peer review history of their article (what does this mean?). If published, this will include your full peer review and any attached files.

Reviewer #1: No

Reviewer #2: No

Reviewer #3: No

---

## [Author Response · Author response to Decision Letter 0]

11 Sep 2022

Please see the attached file as "response to the reviewers".

---

## [Decision Letter · Decision Letter 1]

8 Nov 2022

Structural filtering of functional data offered discriminative features for autism spectrum disorder

PONE-D-22-18416R1

Dear Dr. Mohammadzadeh Asl,

We’re pleased to inform you that your manuscript has been judged scientifically suitable for publication and will be formally accepted for publication once it meets all outstanding technical requirements.

Kind regards,

Moo K. Chung

Academic Editor

PLOS ONE

Additional Editor Comments (optional):

Reviewers' comments:

Reviewer's Responses to Questions

**Comments to the Author**

1. If the authors have adequately addressed your comments raised in a previous round of review and you feel that this manuscript is now acceptable for publication, you may indicate that here to bypass the “Comments to the Author” section, enter your conflict of interest statement in the “Confidential to Editor” section, and submit your "Accept" recommendation.

Reviewer #1: All comments have been addressed

Reviewer #2: (No Response)

2. Is the manuscript technically sound, and do the data support the conclusions?

Reviewer #1: Yes

Reviewer #2: (No Response)

3. Has the statistical analysis been performed appropriately and rigorously? 

Reviewer #1: Yes

Reviewer #2: (No Response)

4. Have the authors made all data underlying the findings in their manuscript fully available?

Reviewer #1: Yes

Reviewer #2: (No Response)

5. Is the manuscript presented in an intelligible fashion and written in standard English?

Reviewer #1: Yes

Reviewer #2: (No Response)

6. Review Comments to the Author

Reviewer #1: Report on revised manuscript

The authors have significantly addressed the comments.

• Overall, this is a clear, concise, and well-written manuscript.

• The introduction is relevant, and theory based.

• Sufficient information about the previous study findings is presented for readers to follow the present study rationale and procedures.

• The methods are generally appropriate.

• Outstanding study and well done.

• Recommendation

Care has been taken to improve the work. I can recommend the acceptance of the manuscript.

Reviewer #2: (No Response)

7. PLOS authors have the option to publish the peer review history of their article (what does this mean?). If published, this will include your full peer review and any attached files.

Reviewer #1: No

Reviewer #2: No

---

## [Editor Report · Acceptance letter]

23 Nov 2022

PONE-D-22-18416R1 

Structural filtering of functional data offered discriminative features for autism spectrum disorder 

Dear Dr. Mohammadzadeh Asl:

I'm pleased to inform you that your manuscript has been deemed suitable for publication in PLOS ONE. Congratulations! Your manuscript is now with our production department. 

Kind regards, 

on behalf of

Dr. Moo K. Chung 

Academic Editor

PLOS ONE